# An exact local mapping from clock-spins to fermions

Simone Traverso[1*], Christoph Fleckenstein[2],
Maura Sassetti[1,3] and Niccolò Traverso Ziani[1,3]

**1** Dipartimento di Fisica, Università degli studi di Genova, 16146 Genova, Italy
**2** KTH Royal Institute of Technology, 106 91 Stockholm, Sweden
**3** SPIN-CNR, 16146 Genova, Italy

★ simone.traverso@edu.unige.it

## Abstract

Clock-spin models are attracting great interest, due to both their rich phase diagram and their connection to parafermions. In this context, we derive an exact local mapping from clock-spin to fermionic partition functions. Such mapping, akin to techniques introduced by Fedotov and Popov for spin $\frac{1}{2}$ chains, grants access to well established numerical tools for the perturbative treatment of fermionic systems in the clock-spin framework. Moreover, in one dimension, it allows to use bosonization to access the low energy properties of clock-spin models. Finally, aside from the direct application in clock-spin models, this new mapping enables the conception of interesting fermionic models, based on the clock-spin counterparts.

# 1  Introduction

Mappings– exact or approximate –are essential tools for the understanding of interacting many body systems. Due to the fact that most often the statistics of the low energy excitations of a physical system does not correspond to that of its elementary constituents, such mappings usually connect operators satisfying different algebras. This fact may be particularly useful when dealing with spin systems. Indeed, even when the mapping does not provide an exact solution, representing spins in terms of fermions and bosons naturally gives access to Wick's theorem and so to all the well established tools based on diagrammatic expansions [1–4].

    A prominent example of mapping between spins and bosons is the Holstein-Primakoff transformations [5], that has been used to effectively describe the emergence of magnons and their interactions [6]. The mappings between spins and fermions define an even richer playground: different schemes can be applied and selected to simplify a given problem. The first of its kind and most widely known was formulated in 1928 by Jordan and Wigner [7], recognizing that spin-1/2 particles possess the same Hilbert space dimension as a single fermionic state. However, to ensure off-site fermionic anticommutation relations they had to introduce string operators rendering the mapping genuinely non-local. The Jordan-Wigner transformation led to countless, important analytical results in models related to the quantum Ising chain [8–15], even out of equilibrium [16–19]. As a source of complexity, however, the intrinsic non locality of the mapping can make the treatment of correlation functions between points that are far away from each other, and the analysis of systems with long range interactions, rather involved. A way out was later provided by Fedotov and Popov [20], who took a different approach: Instead of aligning the Hilbert space dimension, they first focused on ensuring the proper commutation relations by associating two species of fermions to each spin-1/2 operator. Obviously, only a subspace of the fermionic system – dubbed as physical – can then represent the original model and excessive states have to be projected out. Crucially, the Fedotov-Popov approach preserves the locality of the Hamiltonian even in fermionic language, thus making it extremely useful in perturbative methods such as Diagrammatic Monte Carlo simulations. Notably, several extensions of the Fedotov-Popov mapping have been proposed to generalize the scheme to higher spins [21, 22].

    An interesting close relative of spin models, is represented by the so-called *clock-spin* models [23–31]. In contrast to usual spin operators, clock-spin operators obey a generalized Clifford algebra, where on-site commutation of clock-spins acquires a phase factor. This inherent property gives clock-spins a genuine relation to (non-Abelian) anyons with the same on-site commutation relations. The compelling off-site commutation relations present for anyonic particles can again be introduced by virtue of Jordan-Wigner strings [29, 32], as first shown by Fradkin and Kadanoff [23]. Just as clock-spin models present a generalization of the quantum Ising model, the corresponding anyonic or parafermionic models can be interpreted as generalizations of the Kitaev chain [33], where the topological phase is characterised by unpaired

parafermionic modes exponentially localized at the boundaries [34, 35]. Such modes are not only interesting from a fundamental perspective but also due to their potential applications in topological quantum computing [35, 36]. In addition to their relation to parafermions, clock-spin models have been demonstrated to be able to show many-body localization [37] and time-crystalline behavior [37, 38]. Moreover, their phase diagram is extremely rich [39–45]– much richer than the phase diagram of the Ising chain –as gapless incommensurate phases and even phase transitions [46] of the Kosterlitz-Thouless type can take place [47, 48]. Finally, from the experimental point of view, Rydberg atom chains have been shown to possess phase transitions in the same universality class as the ones characterizing clock-spin models [49].

In this work, we develop a mapping that transforms clock-spins directly to fermions. The mapping, which is exact and local, borrows ideas from the Fedotov-Popov transformation. We provide two versions of the mapping, which differ from each other for the choice of the fermionic subspace replicating the clock-spin space. We discuss the implications of these choices and present explicit examples for both cases. Importantly, the mapping ensures identical thermodynamic properties of the Hamiltonians on either end of the transformation. This opens the possibility of studying clock-spin models as well as their anyonic counterparts in a full fermionic language, with the use of established numerical tools, and, in one dimension, of bosonization to assess the low energy physics. The usefulness of the mapping, however, goes beyond numerical computations. With an explicit example we demonstrate that, starting from a known clock-spin model that exhibits exotic ground-state properties, the mapping can also be used in an approximate way to find fermionic models manifesting intriguing behaviour.

The paper is organized as follows: in section 2 we introduce the clock-spin algebra and discuss the general traits of the mapping, recalling the Fedotov-Popov theory and extending it to the case at hand. In section 3 we present a first version of the mapping, characterized by a minimal physical subspace; we give explicit examples of the full mapping for the $n = 3$ and $n = 4$ cases. In section 4 we present a second version for the mapping, characterized by an enlarged physical sector. In this case, we provide a full explicit expression of the mapping for any odd $n$, complemented with the concrete example for $n = 3$. In section 5 we investigate a clock-spin model with exotic features, that – by virtue of our mapping – can be translated to properties correspondingly found in a more realistic fermionic model. Finally, in section 6 we draw our conclusions.

## 2 General traits of the clock-spin to fermions mapping

We consider a generic $d$-dimensional lattice $\Omega$ of $n$-th order clock-spins including a total number of $N$ sites. Moreover, we denote the clock-spin operators at site $j$ as $\sigma_j$ and $\tau_j$. These operators commute at different lattice sites, while on the same site they satisfy the following commutation relations

$$\sigma_j \tau_j = \omega \tau_j \sigma_j, \qquad \sigma_j^n = \tau_j^n = 1, \qquad \sigma_j^\dagger = \sigma_j^{n-1}, \qquad \tau_j^\dagger = \tau_j^{n-1}, \tag{1}$$

with $\omega = \exp\left(i\frac{2\pi}{n}\right)$. For a single site, an explicit representation of the $n$-th order clock-spin operators in a basis in which $\sigma$ is diagonal is given by

$$\sigma = \begin{bmatrix} 1 & 0 & 0 & \cdots & 0 \\ 0 & \omega & 0 & \cdots & 0 \\ 0 & 0 & \omega^2 & \ddots & \vdots \\ \vdots & \vdots & \ddots & \ddots & 0 \\ 0 & 0 & \cdots & 0 & \omega^{n-1} \end{bmatrix}, \qquad \tau = \begin{bmatrix} 0 & 0 & \cdots & 0 & 1 \\ 1 & 0 & \cdots & 0 & 0 \\ 0 & 1 & 0 & \cdots & 0 \\ \vdots & \ddots & \ddots & \ddots & \vdots \\ 0 & \cdots & 0 & 1 & 0 \end{bmatrix}. \tag{2}$$

Note that clock-spin operators of order $n = 2$, $\sigma$ and $\tau$, correspond to the well known Pauli matrices $\sigma_z$ and $\sigma_x$. In this sense the clock-spin operators are a generalization of the spin-$\frac{1}{2}$ operators and the associated Lie-algebra. A renowned example in this respect is the generalization of the $\mathbb{Z}_2$ symmetric Ising model, which becomes a $\mathbb{Z}_n$ symmetric clock-spin model [29].

The goal of the present paper is to transfer the ideas of the Fedotov-Popov theory [20] for spins to the context of generic order clock-spins. In other words, we aim to develop a local mapping under which clock-spin operators transform into fermionic ones while maintaining the partition function, *i.e.* identical thermodynamic properties. Our mapping can be applied to an arbitrary clock-spin model. In the following we will generically write the Hamiltonian as a function of the clock-spin operators at each site of the lattice

$$H^{(n)} = H^{(n)}(\{\sigma_i\}_{i \in \Omega}, \{\tau_i\}_{i \in \Omega}). \tag{3}$$

For the sake of simplicity, we first consider single-site clock-spin operators. The starting point is to map the $n$-th order clock-spin operators $\sigma$ and $\tau$ onto an appropriate combination of $n$ fermionic creation and destruction operators. For the remainder of this section, they are denoted as $f_\alpha^\dagger$ and $f_\alpha$ respectively ($\alpha = 1, \dots, n$). They satisfy the usual canonical anticommutation relations for fermions: $\{f_\alpha^\dagger, f_{\alpha'}\} = \delta_{\alpha, \alpha'}$ and $\{f_\alpha, f_{\alpha'}\} = 0$. In the following we refer to the greek letter identifying the different fermions on each lattice site as the fermionic flavor. For later convenience, we can extend the range of the flavor index $\alpha$ to generic positive integers, with the prescription that the fermionic species are identified by $\alpha \mod n$, *e.g.* $f_{n+1}$ destroys to the same fermion as $f_1$.

The fermionic Hilbert space of $n$ fermions possesses a dimension of $2^n$, whereas the original clock-spin space has dimension $n$. In order to achieve a valid mapping it is necessary to take care of the $2^n - n$[1] fermionic excess states with no corresponding clock-spin state. One way to do so is to label some fermionic states as physical and the remaining ones as unphysical, and deal with the latter in such a way that they do not contribute in the computation of physical quantities.

The choice of the fermionic states used to form the physical subspace is a crucial step in the development of the mapping. For example, a natural choice may be to consider as physical the $n$ states with occupation number ($\hat{N} = \sum_{\alpha=1}^n f_\alpha^\dagger f_\alpha$) equal to one. However, as will be shown below, this is not the only way to proceed. For reasons that will become clear in the subsequent sections, we require that the mapping has the following generic features:

(i) The physical subspace can be divided in an integer number $\ell$ of subsets, each containing $n$ states and independently mapping to the whole clock-spin space.

(ii) The representation of the clock-spin operators in terms of fermions, that we denote here as $\tilde{\sigma}$ and $\tilde{\tau}$, live on a Hilbert space of dimension $2^n$. They act on the physical states just as the clock-spin operators act on the corresponding clock-spin states, while giving zero when acting on the unphysical states.

(iii) $\tilde{\sigma}$ and $\tilde{\tau}$ only contain terms which are product of an even number of fermionic creation/destruction operators.

Once the correct mapping for the $\sigma$ and $\tau$ operators on a single site has been established, it can be extended identically to any site. This is done by defining fermionic destruction and creation operators $f_{j,\alpha}$ and $f_{j,\alpha}^\dagger$ on every site of the lattice $\Omega$. Then, $\tilde{\sigma}_j$ and $\tilde{\tau}_j$ are found by replacing the fermionic operators in the single site expressions (which satisfy the conditions (i), (ii) and (iii)) with those at site $j$. In particular, condition (iii) ensures that $[\tilde{\sigma}_i, \tilde{\tau}_j] = 0$ for

---

[1]In this work, we discuss two valid versions of the mapping that deal with a different number of fermionic excess states: $2^n - n$ and $2^n - 2n$, respectively.

$i \neq j$. This enables us to write down the mapping from the clock-spin Hamiltonian (3) to the corresponding fermionic one:

$$H^{(n)} \mapsto H_{\mathrm{F}}^{(n)} = H^{(n)}(\{\tilde{\sigma}_i\}_{i \in \Omega}, \{\tilde{\tau}_i\}_{i \in \Omega}). \tag{4}$$

By construction, the action of $H_{\mathrm{F}}^{(n)}$ on the physical states – that for the lattice are defined as the tensor product of physical states on every site – is the same as that of $H^{(n)}$ on the corresponding clock-spin states. Then, the partition function corresponding to the clock-spin Hamiltonian, that we denote as $Z_{\mathrm{c.s.}}$, can be equivalently computed by taking the trace of $e^{-\beta H_{\mathrm{F}}^{(n)}}$ just on the physical states

$$\mathrm{Tr}_{\mathrm{c.s.}} \, e^{-\beta H^{(n)}} = \mathrm{Tr}_{\mathrm{phys}} \, e^{-\beta H_{\mathrm{F}}^{(n)}}. \tag{5}$$

The right and left side of Eq. (5) are identical up to an integer factor ($\ell^N$) accounting for the ratio between the dimension of the physical subspace and that of the actual clock-space (if different from unity).

Following Ref. [20], the next step consists of extending the trace over the whole fermionic Hilbert space while preserving the above identification. For that, we need to add a suitable imaginary interaction term $iO_{\mathrm{F}}^{(n)}$ to the Hamiltonian, such that the trace over the unphysical states does not contribute to the final result

$$\mathrm{Tr}_{\mathrm{c.s.}} \, e^{-\beta H^{(n)}} = \mathrm{Tr} \, e^{-\beta(H_{\mathrm{F}}^{(n)} + iO_{\mathrm{F}}^{(n)})}. \tag{6}$$

Suppose that the imaginary interaction term has the same form on each site, so that it can be decomposed as

$$O_{\mathrm{F}}^{(n)} = \sum_{j=1}^{N} O_{\mathrm{F}_j}^{(n)}, \tag{7}$$

with each of the $O_{\mathrm{F}_j}^{(n)}$ terms involving just operators on site $j$. Furthermore, assume that $O_{\mathrm{F}_j}^{(n)}$ consists only of combinations of fermionic number operators, *i.e.* terms of the form $f_{j\alpha}^{\dagger} f_{j\alpha}$. Now let us focus on the following family of states defined on the whole lattice

$$|s\rangle = |\mathrm{unph}_i\rangle \{\otimes_{j \neq i} |s_j\rangle\}, \tag{8}$$

that present an unphysical state at site $i$ and any state, either physical or unphysical, at the other sites. We want to compute the trace of the operator $e^{-\beta(H_{\mathrm{F}}^{(n)} + iO_{\mathrm{F}}^{(n)})}$ over the different unphysical states at site $i$, while leaving the part of the state concerning the other sites unchanged. That is

$$\sum_{\mathrm{unph}_i} \{\otimes_{j \neq i} \langle s_j|\} \langle \mathrm{unph}_i| \, e^{-\beta(H_{\mathrm{F}}^{(n)} + iO_{\mathrm{F}}^{(n)})} \, |\mathrm{unph}_i\rangle \{\otimes_{j \neq i} |s_j\rangle\}. \tag{9}$$

In order to simplify the notation, from here until the end of this section we omit the $(n)$ superscript on the operators. Let us denote with $H_{\mathrm{F}_i}$ ($O_{\mathrm{F}_i}$) the part of $H_{\mathrm{F}}$ ($O_{\mathrm{F}}$) that collects all the fermionic operators at site $i$ and with $H_{\mathrm{F}_i}'$ ($O_{\mathrm{F}_i}'$) the remaining part of the operator. Now, if the condition

$$H_{\mathrm{F}_i} |\mathrm{unph}_i\rangle = 0 \tag{10}$$

is fulfilled– which is indeed the case if $\tilde{\sigma}_i |\mathrm{unph}_i\rangle = 0 = \tilde{\tau}_i |\mathrm{unph}_i\rangle$ as required from condition (ii) –then Eq. (9) reduces to

$$\{\otimes_{j \neq i} \langle s_j|\} e^{-\beta(H_{\mathrm{F}_i}' + iO_{\mathrm{F}_i}')} \{\otimes_{j \neq i} |s_j\rangle\} \sum_{|\mathrm{unph}_i\rangle} \langle \mathrm{unph}_i| \, e^{-i\beta O_{\mathrm{F}_i}} \, |\mathrm{unph}_i\rangle, \tag{11}$$

as demonstrated in appendix A. Thus, to ensure that the unphysical states do not contribute to the trace, one has to choose the form of $O_{F_i}$ such that

$$\sum_{|\text{unph}_i\rangle} \langle \text{unph}_i | e^{-i\beta O_{F_i}} | \text{unph}_i \rangle = 0 \,. \tag{12}$$

Indeed, if condition (12) is satisfied then

$$\text{Tr}\, e^{-\beta(H_F + iO_F)} = \text{Tr}_{\text{phys}}\, e^{-\beta(H_F + iO_F)} \,. \tag{13}$$

The remaining last step consists in relating the trace over the physical states of $e^{-\beta(H_F + iO_F)}$ to the clock-spin partition function. However, this identification strictly depends on the choice made about the physical subspace and on the explicit form of $O_F$.

Now that the general framework of the theory has been set up, we proceed with the actual development of the mapping. The task will be addressed in two distinct ways, that differ for the choice of the physical subspace at each site of the lattice: in the next section we choose the $n$ states with occupation number $\hat{N}_j = 1$ to form the physical subspace at site $j$. Instead, in Sec. 4 we build the physical sector from the $n$ states with occupation number $\hat{N}_j = 1$, and those with occupation number $\hat{N}_j = n-1$.

## 3 Minimal physical subspace mapping

In this section we develop the mapping choosing as physical states those with occupation number equal to one. Although this choice makes the implementation straightforward, it also induces a major issue: With this version of the mapping it is not possible to provide a general expression for the imaginary interaction term $iO_F^{(n)}$ as a function of the clock-spin order $n$. Here we will focus on the derivation of its expression just for the two examples of $n = 3$ and $n = 4$. Though similar procedures could be followed for higher values of $n$, a closed formula cannot be given.

Let us first consider a single site. After all, as outlined in Sec. 2, once the single site operators are mapped to fermions, the extension of the mapping to the whole lattice is trivial. We denote the physical states as $f_\alpha^\dagger |0\rangle$, $\alpha \in \{1, \dots, n\}$. As a starting point, we follow Abrikosov's proposal [50] to map spin $\frac{1}{2}$ operators to fermions, and write

$$\tilde{\sigma}_A = \sum_{\alpha,\beta=1}^{n} f_\alpha^\dagger \sigma_{\alpha\beta} f_\beta \,, \tag{14}$$

$$\tilde{\tau}_A = \sum_{\alpha,\beta=1}^{n} f_\alpha^\dagger \tau_{\alpha\beta} f_\beta \,, \tag{15}$$

with $\sigma_{\alpha\beta}$ and $\tau_{\alpha\beta}$ the matrices defined in Eq. (2). Then, the action of $\tilde{\sigma}_A$ and $\tilde{\tau}_A$ on the physical states is given by

$$\tilde{\sigma}_A f_\alpha^\dagger |0\rangle = \omega^{\alpha-1} f_\alpha^\dagger |0\rangle \,, \tag{16}$$

$$\tilde{\tau}_A f_\alpha^\dagger |0\rangle = f_{\alpha+1}^\dagger |0\rangle \,, \tag{17}$$

where the identification $f_{n+1}^\dagger = f_1^\dagger$ is implied in the second line. From Eq. (16) and Eq. (17), it seems to be well justified to identify the eigenstate with eigenvalue $\omega^{\alpha-1}$ of $\sigma$ with the fermionic state $f_\alpha^\dagger |0\rangle$. However, a problem arises when we consider the action of these operators on the states that we labeled as *unphysical*. It is easy to see that the action of both $\tilde{\sigma}_A$ and

$\tilde{\tau}_A$ on these states is nor zero – except for those with occupation number $\hat{N} = 0$ and $\hat{N} = n$ – neither the same as on the states we dubbed as physical. In order to achieve the desired action on all the unphysical states, we modify the above expressions of $\tilde{\sigma}_A$ and $\tilde{\tau}_A$ in the following way

$$\tilde{\sigma} = \sum_{\alpha=1}^{n} \omega^{\alpha-1} \hat{n}_\alpha \prod_{\substack{\rho=1 \\ \rho \neq \alpha}}^{n} (1 - \hat{n}_\rho), \tag{18}$$

$$\tilde{\tau} = \sum_{\alpha=1}^{n} f_{\alpha+1}^\dagger f_\alpha \prod_{\substack{\rho=1 \\ \rho \neq \alpha, \alpha+1}}^{n} (1 - \hat{n}_\rho), \tag{19}$$

where $\hat{n}_\alpha = f_\alpha^\dagger f_\alpha$ is the occupation number operator associated to the flavor $\alpha$. One can see that these new definitions leave the action of $\tilde{\sigma}$ and $\tilde{\tau}$ unchanged in the physical subspace. However, the introduction of the products of terms $(1 - \hat{n}_\rho)$ assures that they yield zero when acting on any fermionic state with occupation number different from one. The fermionic operators in Eq. (18) and Eq. (19) correctly reproduce the commutation relations of their clock-spin counterparts (see Eq. (1)). This fact is most easily verified by computing and comparing the action of $\tilde{\sigma}\tilde{\tau}$ and $\tilde{\tau}\tilde{\sigma}$ on a complete basis of fermionic states. For what it concerns the physical sector $\tilde{\sigma}$ and $\tilde{\tau}$ act on the fermionic states as $\sigma$ and $\tau$ act on the corresponding clock-spin states, so that the correct algebra is automatically implemented. Moreover, also in the unphysical sector the commutation relations are trivially satisfied thanks to the fact that $\tilde{\sigma}$ and $\tilde{\tau}$ are zero on the whole subspace.

We can now extend the mapping to the whole lattice, by associating $n$ fermionic operators to each site. Then, the operators at site $j$ will be written in terms of fermionic operators at site $j$ only, with the same functional dependence derived for the single site

$$\tilde{\sigma}_j = \sum_{\alpha=1}^{n} \omega^{\alpha-1} \hat{n}_{j,\alpha} \prod_{\substack{\rho=1 \\ \rho \neq \alpha}}^{n} (1 - \hat{n}_{j,\rho}), \tag{20}$$

$$\tilde{\tau}_j = \sum_{\alpha=1}^{n} f_{j,\alpha+1}^\dagger f_{j,\alpha} \prod_{\substack{\rho=1 \\ \rho \neq \alpha, \alpha+1}}^{n} (1 - \hat{n}_{j,\rho}). \tag{21}$$

We note that this implementation of the mapping automatically ensures the condition

$$H_{F_j}^{(n)} |\text{unph}_j\rangle = 0. \tag{22}$$

Moreover, since $\tilde{\sigma}_j$ and $\tilde{\tau}_j$ only contain terms which are the product of an even number of fermionic operators at site $j$, condition (iii) is naturally satisfied. Then we have

$$[\tilde{\sigma}_i, \tilde{\tau}_j] = [\tilde{\tau}_i, \tilde{\tau}_j] = [\tilde{\sigma}_i, \tilde{\sigma}_j] = 0,$$

for any $i \neq j$. Henceforth, the mapped fermionic operators correctly reproduce the off-site commutation relations of the clock-spin operators.

In the remainder of this section we use Eqs. (20) and (21) to explicitly derive the mapping for the cases $n = 3$ and $n = 4$. While working out these examples, we also derive appropriate expressions for the imaginary interaction term $iO_F^{(n)}$, proceeding case by case. By doing so, it becomes clear what hinders the derivation of a closed formula for a generic order $n$, at least inside the framework provided by the present mapping. For the case of odd order clock-spins this problem is overcome in the next section, where we develop the mapping starting from a doubled physical subspace.

## 3.1 The $n = 3$ case

As a first example, we consider the $n = 3$ case. To simplify the notation, in this subsection we denote the annihilation operators associated with the three fermions involved in the mapping for each site $j$ as $a_j$, $b_j$, $c_j$ instead of $f_{j,1}$, $f_{j,2}$, $f_{j,3}$.

Before deriving the full expression of the correct mapping according to Eqs. (20) and (21), it can be interesting to focus on a single site again and try to consider the Abrikosov-like implementation, that is

$$\tilde{\sigma}_A = a^\dagger a + \omega b^\dagger b + \omega^2 c^\dagger c, \tag{23}$$

$$\tilde{\tau}_A = b^\dagger a + c^\dagger b + a^\dagger c, \tag{24}$$

where, being $n = 3$, $\omega = e^{i\frac{2\pi}{3}}$. As outlined in former sections, these operators correctly reproduce the action of the clock-spin operators on the clock-spin states, as far as the physical subspace is concerned. However, the action on the unphysical states is given by

$$\tilde{\sigma}_A |0, 0, 0\rangle = 0, \qquad\qquad\qquad \tilde{\tau}_A |0, 0, 0\rangle = 0, \tag{25}$$

$$\tilde{\sigma}_A |1, 1, 0\rangle = (1 + \omega) |1, 1, 0\rangle, \qquad \tilde{\tau}_A |1, 1, 0\rangle = |1, 0, 1\rangle, \tag{26}$$

$$\tilde{\sigma}_A |1, 0, 1\rangle = (1 + \omega^2) |1, 0, 1\rangle, \qquad \tilde{\tau}_A |1, 0, 1\rangle = |0, 1, 1\rangle, \tag{27}$$

$$\tilde{\sigma}_A |0, 1, 1\rangle = (\omega + \omega^2) |0, 1, 1\rangle, \qquad \tilde{\tau}_A |0, 1, 1\rangle = |1, 1, 0\rangle, \tag{28}$$

$$\tilde{\sigma}_A |1, 1, 1\rangle = (1 + \omega + \omega^2) |1, 1, 1\rangle = 0, \qquad \tilde{\tau}_A |1, 1, 1\rangle = 0. \tag{29}$$

Here we see explicitly what we asserted at the beginning of this section: if one implements the mapping as in Eqs. (14) and (15), then the action on some of the unphysical states– those with occupation number $\hat{N} = 2$ for the present example –does not yield zero, nor the same as on the physical states. This justifies the mapping in Eqs. (18) and (19), which in some sense projects the states onto the physical subspace. With this in mind, we can proceed to use Eqs. (20) and (21) to write down the complete mapping of the clock-spin operators for the whole lattice. Shifting to the present notation and performing some minor algebraic simplifications we find

$$\tilde{\sigma}_j = a_j^\dagger a_j (1 - b_j^\dagger b_j - c_j^\dagger c_j) + \omega b_j^\dagger b_j (1 - a_j^\dagger a_j - c_j^\dagger c_j) + \omega^2 c_j^\dagger c_j (1 - a_j^\dagger a_j - b_j^\dagger b_j), \tag{30}$$

$$\tilde{\tau}_j = a_j^\dagger c_j (1 - b_j^\dagger b_j) + b_j^\dagger a_j (1 - c_j^\dagger c_j) + c_j^\dagger b_j (1 - a_j^\dagger a_j). \tag{31}$$

By plugging these expressions into any $H^{(3)}$ clock-spin Hamiltonian akin to Eq. (4), we obtain the corresponding fermionic Hamiltonian, $H_F^{(3)}$.

Next, we start looking for a suitable imaginary interaction term. Recall that we require it to be decomposable as $O_F^{(3)} = \sum_{j=1}^{N} O_{F_j}^{(3)}$, with each of the $O_{F_j}^{(3)}$ having the same functional form and being built from fermionic number operators at site $j$ only. Moreover, $O_{F_j}^{(3)}$ must satisfy Eq. (12) for any site index $j$.

Suppose first that, for the sake of simplicity, we try to take $O_{F_j}^{(3)}$ symmetric for the exchange of any two fermionic operators at site $j$, meaning that

$$O_{F_j}^{(3)}\left(\{f_{j,\alpha}\}, \{f_{j,\alpha}^\dagger\}\right) = O_{F_j}^{(3)}\left(\{f_{j,p(\alpha)}\}, \{f_{j,p(\alpha)}^\dagger\}\right), \tag{32}$$

for any permutation $p$ of the flavor indices. If this condition holds, then clearly $O_{F_j}^{(3)}$ will have to be degenerate on any subspace with fixed fermionic on-site number $\hat{N}_j = \sum_{\alpha=1}^{n} f_{j,\alpha}^\dagger f_{j,\alpha}$. If we introduce a set of coefficients $\eta_k, k \in \{0, \dots, n\}$, such that

$$O_{F_j}^{(3)} |\hat{N}_j = k\rangle = \beta^{-1} \eta_k |\hat{N}_j = k\rangle, \tag{33}$$

then Eq. (12) becomes

$$\sum_{|\mathrm{unph}_j\rangle} \langle\mathrm{unph}_j| e^{-i\beta O_{\mathrm{F}_j}^{(3)}} |\mathrm{unph}_j\rangle = \binom{3}{0}e^{-i\eta_0} + \binom{3}{2}e^{-i\eta_2} + \binom{3}{3}e^{-i\eta_3} = 0. \tag{34}$$

Unfortunately, a set of parameters $\eta_\ell$ for which the above sum yields zero cannot be found. This can be easily understood by taking the norm of the sum and applying the triangular inequality. The only way to satisfy Eq. (12) is then to pick an operator $O_{\mathrm{F}_j}^{(3)}$ that is asymmetric in fermionic operators. Choosing

$$O_{\mathrm{F}_j}^{(3)} = \frac{\pi}{3\beta}(a_j^\dagger a_j(1 + 2c_j^\dagger c_j) + b_j^\dagger b_j(1 + 2a_j^\dagger a_j) + c_j^\dagger c_j(1 + 2a_j^\dagger a_j)), \tag{35}$$

we have

$$O_{\mathrm{F}_j}^{(3)} |0,0,0\rangle_j = 0, \tag{36}$$

$$O_{\mathrm{F}_j}^{(3)} |1,1,0\rangle_j = \frac{4\pi}{3\beta} |1,1,0\rangle_j, \tag{37}$$

$$O_{\mathrm{F}_j}^{(3)} |1,0,1\rangle_j = \frac{6\pi}{3\beta} |1,0,1\rangle_j, \tag{38}$$

$$O_{\mathrm{F}_j}^{(3)} |0,1,1\rangle_j = \frac{2\pi}{3\beta} |0,1,1\rangle_j, \tag{39}$$

$$O_{\mathrm{F}_j}^{(3)} |1,1,1\rangle_j = \frac{9\pi}{3\beta} |1,1,1\rangle_j, \tag{40}$$

and, thus, Eq. (12) is satisfied:

$$\sum_{|\mathrm{unph}_j\rangle} \langle\mathrm{unph}_j| e^{-i\beta O_{\mathrm{F}_j}^{(3)}} |\mathrm{unph}_j\rangle = 1 + e^{-i\frac{4\pi}{3}} + e^{-i\frac{6\pi}{3}} + e^{-i\frac{2\pi}{3}} + e^{-i\frac{9\pi}{3}} = 0. \tag{41}$$

Of course the choice of $O_{\mathrm{F}_j}^{(3)}$ is not unique: different asymmetric implementations would still give the desired result, as long as the condition of the trace over the unphysical sector being zero is satisfied. To conclude, we observe that the action of $O_{\mathrm{F}_j}^{(3)}$ is the same on any of the physical states at site $j$,

$$O_{\mathrm{F}_j}^{(3)} |\mathrm{phys}_j\rangle = \frac{\pi}{3\beta} |\mathrm{phys}_j\rangle. \tag{42}$$

Then, as the last step, we redefine the total $O_{\mathrm{F}}^{(3)}$ operator as follows

$$O_{\mathrm{F}}^{(3)} = \sum_{j=1}^{N} \frac{\pi}{3\beta}[(a_j^\dagger a_j(1 + 2c_j^\dagger c_j) + b_j^\dagger b_j(1 + 2a_j^\dagger a_j) + c_j^\dagger c_j(1 + 2a_j^\dagger a_j))] - \frac{\pi}{3\beta}N, \tag{43}$$

so that its action on the physical states yields zero. Crucially, adding a constant term to $O_{\mathrm{F}}^{(3)}$ has no effect on the trace over the unphysical states. Indeed, if we replace $O_{\mathrm{F}_j}^{(3)}$ by $O_{\mathrm{F}_j}^{(3)} + C$ in Eq. (41) ($C \in \mathbb{R}$) we get

$$\sum_{|\mathrm{unph}_j\rangle} \langle\mathrm{unph}_j| e^{-i\beta(O_{\mathrm{F}_j}^{(3)} + C)} |\mathrm{unph}_j\rangle = e^{-i\beta C} \sum_{|\mathrm{unph}_j\rangle} \langle\mathrm{unph}_j| e^{-i\beta O_{\mathrm{F}_j}^{(3)}} |\mathrm{unph}_j\rangle = 0, \tag{44}$$

so that the trace over the unphysical states still add up to zero.

Eventually, we obtain

$$\mathrm{Tr}\, e^{-\beta(H_{\mathrm{F}}^{(3)} + iO_{\mathrm{F}}^{(3)})} = \mathrm{Tr}_{\mathrm{phys}}\, e^{-\beta(H_{\mathrm{F}}^{(3)} + iO_{\mathrm{F}}^{(3)})} = \mathrm{Tr}_{\mathrm{phys}}\, e^{-\beta H_{\mathrm{F}}^{(3)}} = \mathrm{Tr}_{\mathrm{c.s.}}\, e^{-\beta H^{(3)}}. \tag{45}$$

The partition function corresponding to the original clock-spin Hamiltonian is just the same as the one corresponding to the non-Hermitian fermionic Hamiltonian $H_{\mathrm{F}}^{(3)} + iO_{\mathrm{F}}^{(3)}$.

## 3.2 The $n = 4$ case

For $n = 4$ we can proceed along the same lines as for $n = 3$, except for one notable difference: in this case the single-site interaction term $O_{F_j}^{(4)}$ can be chosen symmetric for the exchange of the fermionic operators, in the sense of Eq. (32).

In analogy to what was done in the previous example, we will denote the annihilation operators associated to the four fermions involved in the mapping (for each site $j$) as $a_j$, $b_j$, $c_j$, $d_j$ respectively. According to Eq. (20), the $\sigma$ operator at site $j$ ($\sigma_j$) is mapped into

$$
\begin{aligned}
\tilde{\sigma}_j = {} & a_j^\dagger a_j (1 - b_j^\dagger b_j - c_j^\dagger c_j - d_j^\dagger d_j + b_j^\dagger b_j c_j^\dagger c_j + b_j^\dagger b_j d_j^\dagger d_j + c_j^\dagger c_j d_j^\dagger d_j) \\
& + \omega b_j^\dagger b_j (1 - a_j^\dagger a_j - c_j^\dagger c_j - d_j^\dagger d_j + a_j^\dagger a_j c_j^\dagger c_j + a_j^\dagger a_j d_j^\dagger d_j + c_j^\dagger c_j d_j^\dagger d_j) \\
& + \omega^2 c_j^\dagger c_j (1 - b_j^\dagger b_j - a_j^\dagger a_j - d_j^\dagger d_j + b_j^\dagger b_j a_j^\dagger a_j + b_j^\dagger b_j d_j^\dagger d_j + a_j^\dagger a_j d_j^\dagger d_j) \\
& + \omega^3 d_j^\dagger d_j (1 - b_j^\dagger b_j - c_j^\dagger c_j - a_j^\dagger a_j + b_j^\dagger b_j c_j^\dagger c_j + b_j^\dagger b_j a_j^\dagger a_j + c_j^\dagger c_j a_j^\dagger a_j),
\end{aligned}
\tag{46}
$$

where in this case $\omega = i$. Furthermore, following Eq. (21), $\tau_j$ transforms as

$$
\begin{aligned}
\tilde{\tau}_j = {} & a_j^\dagger d_j (1 - b_j^\dagger b_j - c_j^\dagger c_j + b_j^\dagger b_j c_j^\dagger c_j) \\
& + b_j^\dagger a_j (1 - c_j^\dagger c_j - d_j^\dagger d_j + c_j^\dagger c_j d_j^\dagger d_j) \\
& + c_j^\dagger b_j (1 - a_j^\dagger a_j - d_j^\dagger d_j + a_j^\dagger a_j d_j^\dagger d_j) \\
& + d_j^\dagger c_j (1 - a_j^\dagger a_j - b_j^\dagger b_j + a_j^\dagger a_j b_j^\dagger b_j).
\end{aligned}
\tag{47}
$$

As desired, the operators defined this way act on the physical states at site $j$ ($\hat{N}_j = 1$) as the original clock-spin operators act on the corresponding clock states. On the other hand, they yield zero when applied onto the unphysical states at site $j$ ($\hat{N}_j \in \{0, 2, 3, 4\}$). Next, we determine the operator $O_F^{(4)} = \sum_{j=1}^{N} O_{F_j}^{(4)}$.

As done for the $n = 3$ case, we first look for solutions of Eq. (12) that are symmetric under the exchange of any two fermionic operators at site $j$. Then, leaning on the the same arguments supporting Eq. (33), we write

$$
O_{F_j}^{(4)} |\hat{N}_j = k\rangle = \beta^{-1} \eta_k |\hat{N}_j = k\rangle .
\tag{48}
$$

Consequently, Eq. (12) becomes

$$
\sum_{|\text{unph}_j\rangle} \langle \text{unph}_j | e^{-i\beta O_{F_j}^{(4)}} | \text{unph}_j \rangle = \binom{4}{0} e^{-i\eta_0} + \binom{4}{2} e^{-i\eta_2} + \binom{4}{3} e^{-i\eta_3} + \binom{4}{4} e^{-i\eta_4} = 0 .
\tag{49}
$$

In contrast to $n = 3$, the above equation can indeed be satisfied for a proper choice of the phases. For example, if one takes $\eta_0, \eta_3, \eta_4$ to be integer multiples of $2\pi$ and $\eta_2 = \pi + 2\pi\ell$ with $\ell \in \mathbb{Z}$, then

$$
\sum_{|\text{unph}_j\rangle} \langle \text{unph}_j | e^{-i\beta O_{F_j}^{(4)}} | \text{unph}_j \rangle = 1 - 6 + 4 + 1 = 0 .
\tag{50}
$$

We note that the existence of such an easy solution for the $n = 4$ case is quite peculiar. Indeed for higher (even) values of $n$, solutions of the equations corresponding to Eq. (49) with all the phases equal to $\pm 1$ rarely exist and, in general, possible solutions sensitively depend on $n$.

It is now straightforward to write down a proper candidate for $O_F^{(4)}$. The explicit calculations are done in appendix B; here we simply report two of many possible solutions. An

expression of $O_\text{F}^{(4)}$ that involves one and two-particle interactions only is given by

$$O_\text{F}^{(4)} = \frac{\pi}{\beta} \sum_{j=1}^{N} \Big[ a_j^\dagger a_j + b_j^\dagger b_j + c_j^\dagger c_j + d_j^\dagger d_j$$
$$+ (a_j^\dagger a_j b_j^\dagger b_j + a_j^\dagger a_j c_j^\dagger c_j + a_j^\dagger a_j d_j^\dagger d_j + b_j^\dagger b_j c_j^\dagger c_j + b_j^\dagger b_j d_j^\dagger d_j + c_j^\dagger c_j d_j^\dagger d_j) \Big] - \frac{\pi}{\beta} N . \quad (51)$$

In case one wants to avoid one-particle terms– that give non zero contribution in the physical sector –another possible choice is

$$O_\text{F}^{(4)} = \frac{\pi}{\beta} \sum_{j=1}^{N} (a_j^\dagger a_j b_j^\dagger b_j + a_j^\dagger a_j c_j^\dagger c_j + a_j^\dagger a_j d_j^\dagger d_j + b_j^\dagger b_j c_j^\dagger c_j + b_j^\dagger b_j d_j^\dagger d_j + c_j^\dagger c_j d_j^\dagger d_j)$$
$$+ (a_j^\dagger a_j b_j^\dagger b_j c_j^\dagger c_j + a_j^\dagger a_j b_j^\dagger b_j d_j^\dagger d_j + a_j^\dagger a_j c_j^\dagger c_j d_j^\dagger d_j + b_j^\dagger b_j c_j^\dagger c_j d_j^\dagger d_j) . \quad (52)$$

One can readily check that both these expressions are such that Eq. (6) holds.

## 4 Doubled physical subspace mapping

In this section we provide an alternative implementation of the mapping, which mainly differs from the previous one in the partitioning of physical and unphysical sectors. We will see that for this new version of the mapping, at least in the case of odd clock-spin order $n$, we will be able to write down a generic formula for the imaginary potential term. The price to pay in order to get this appealing feature, which was absent in the first version of the mapping, is that one has to deal with more complicated expressions for the mapped fermionic operators. In order to avoid confusion between the two versions of the mapping, we will denote these latter fermionic representations of the clock-spin operators as $\bar\sigma$ and $\bar\tau$ instead of $\tilde\sigma$ and $\tilde\tau$.

The idea here is to consider as physical both the states with occupation number 1 and $n-1$. Indeed, there are exactly as many states with occupation number $n-1$ as with occupation number 1, that is, $\binom{n}{1} = \binom{n}{n-1} = n$. Using Eqs. (18) and (19) as a starting point, we need to add a second term each with non-zero action on the $\hat N = n-1$ subspace only. These terms are chosen in such a way that the action of $\bar\tau$ and $\bar\sigma$ in this subspace is perfectly equivalent to that in the $\hat N = 1$ subspace. These considerations suggest the following identification for the single site clock-spin operators in terms of fermions

$$\bar\sigma = \sum_{\alpha=1}^{n} \left[ \omega^{\alpha-1} \hat n_\alpha \prod_{\substack{\rho=1 \\ \rho\neq\alpha}}^{n} (1 - \hat n_\rho) + \omega^{n-\alpha}(1 - \hat n_\alpha) \prod_{\substack{\rho=1 \\ \rho\neq\alpha}}^{n} \hat n_\rho \right] , \quad (53)$$

$$\bar\tau = \sum_{\alpha=1}^{n} f_{\alpha+1}^\dagger f_\alpha \left[ \prod_{\substack{\rho=1 \\ \rho\neq\alpha,\alpha+1}}^{n} (1 - \hat n_\rho) + \prod_{\substack{\rho=1 \\ \rho\neq\alpha,\alpha+1}}^{n} \hat n_\rho \right] . \quad (54)$$

It is straightforward to check that the new terms added to $\bar\sigma$ and $\bar\tau$ have non-zero action just on the subspace with occupation number $n-1$. The action on the states with $\hat N = 1$ coincides with that of $\tilde\sigma$ and $\tilde\tau$, which has already been discussed. In the occupation number basis, it can be represented as

$$\bar\tau \, |0,\ldots,0,0,\underbrace{1}_{\alpha},0,0,\ldots,0\rangle = |0,\ldots,0,0,0,\underbrace{1}_{\alpha+1},0,\ldots,0\rangle ,$$
$$\bar\sigma \, |0,\ldots,0,0,\underbrace{1}_{\alpha},0,0,\ldots,0\rangle = \omega^{\alpha-1} |0,\ldots,0,0,\underbrace{1}_{\alpha},0,0,\ldots,0\rangle .$$

On the other hand, the action of $\bar{\tau}$ on a state with occupation number $n-1$ is

$$\bar{\tau}\,|1,\ldots,1,1,\underbrace{0}_{\alpha},1,1,\ldots,1\rangle = |1,\ldots,1,\underbrace{0}_{\alpha-1},1,1,1,\ldots,1\rangle\,.$$

In other words, $\bar{\tau}$ moves the only occupied state in the elements of the subspace $\hat{N}=1$ "to the right", while moving the empty state in the elements of the subspace $\hat{N}=n-1$ "to the left". Instead, the action of $\bar{\sigma}$ just yields a phase

$$\bar{\sigma}\,|1,\ldots,1,1,\underbrace{0}_{\alpha},1,1,\ldots,1\rangle = \omega^{n-\alpha}\,|1,\ldots,1,1,\underbrace{0}_{\alpha},1,1,\ldots,1\rangle\,,$$

$$\bar{\sigma}\,|1,\ldots,1,\underbrace{0}_{\alpha-1},1,1,1,\ldots,1\rangle = \omega^{n-\alpha+1}\,|1,\ldots,1,\underbrace{0}_{\alpha-1},1,1,1,\ldots,1\rangle\,.$$

This justifies the opposite sign of the phases in the second part of $\bar{\sigma}$: it is chosen to ensure that the on-site commutation relations are satisfied on the $\hat{N}=n-1$ subspace too. The extension of the mapping to the whole chain is performed again by defining fermionic operators $f_{j,\alpha}, f_{j,\alpha}^{\dagger}$ for each site to get

$$\bar{\sigma}_j = \sum_{\alpha=1}^{n}\left[\omega^{\alpha-1}\hat{n}_{j,\alpha}\prod_{\substack{\rho=1\\\rho\neq\alpha}}^{n}(1-\hat{n}_{j,\rho}) + \omega^{n-\alpha}(1-\hat{n}_{j,\alpha})\prod_{\substack{\rho=1\\\rho\neq\alpha}}^{n}\hat{n}_{j,\rho}\right]\,, \tag{55}$$

$$\bar{\tau}_j = \sum_{\alpha=1}^{n} f_{j,\alpha+1}^{\dagger}f_{j,\alpha}\left[\prod_{\substack{\rho=1\\\rho\neq\alpha,\alpha+1}}^{n}(1-\hat{n}_{j,\rho}) + \prod_{\substack{\rho=1\\\rho\neq\alpha,\alpha+1}}^{n}\hat{n}_{j,\rho}\right]\,. \tag{56}$$

Next, we derive an explicit expression for the imaginary potential term. The constraints we demand for $O_{\mathrm{F}}^{(n)}$ are the same we discussed in the previous section: we assume that it can be decomposed as a sum of operators $O_{\mathrm{F}_j}^{(n)}$, each of which has the same functional form and only contains number operators at site $j$. Moreover, we require each addendum $O_{\mathrm{F}_j}^{(n)}$ to be invariant under the exchange of any two fermionic operators at site $j$. Similar to Secs. 3.1 and 3.2, we denote

$$O_{\mathrm{F}_j}^{(n)}\,|\hat{N}_j = k\rangle = \beta^{-1}\eta_k\,|\hat{N}_j = k\rangle\,, \tag{57}$$

for any site-$j$ state with occupation number equal to $k$.

Then the condition of Eq. (12) yields

$$\sum_{|\mathrm{unph}_j\rangle}\langle\mathrm{unph}_j|\,e^{-i\beta O_{\mathrm{F}_j}^{(n)}}\,|\mathrm{unph}_j\rangle = \sum_{\substack{k=0\\k\neq 1,n-1}}^{n}\binom{n}{k}e^{-i\eta_k} = 0\,. \tag{58}$$

In the case of odd $n$, we can write $n=2m+1$ and separate the sum as follows

$$\sum_{\substack{k=0\\k\neq 1,n-1}}^{n}\binom{n}{k}e^{-i\eta_k} = \sum_{\substack{k=0\\k\neq 1}}^{m}\binom{2m+1}{k}e^{-i\eta_k} + \sum_{\substack{k=m+1\\k\neq 2m}}^{2m+1}\binom{2m+1}{k}e^{-i\eta_k}\,. \tag{59}$$

Using $\binom{n}{k}=\binom{n}{n-k}$, one can finally rewrite Eq. (59) as

$$\sum_{\substack{k=0\\k\neq 1}}^{m}\binom{2m+1}{k}[e^{-i\eta_k} + e^{-i\eta_{2m+1-k}}] = 0\,. \tag{60}$$

The easiest way to satisfy this condition is to choose the parameters $\eta_k$ in such a way that

$$\eta_k = \eta_{2m+1-k} + (2\ell + 1)\pi, \qquad \ell \in \mathbb{Z}. \tag{61}$$

A trivial solution to Eq. (61) is given by $\eta_k = k\pi$, which corresponds to $O_{F_j}^{(n)} = \beta^{-1}\hat{N}_j\pi$. However, it is necessary that $\eta_1$ and $\eta_{2m}$ do not satisfy the above condition: in that case the contribution to the trace from the physical states with $\hat{N}_j = 1$ would cancel out with the one coming from those with $\hat{N}_j = n-1$. This issue can be solved by adding an appropriate term to $O_{F_j}^{(n)}$, which prevents the cancellation of the physical contributions to the trace. A possible solution is:

$$O_{F_j}^{(n)} = \beta^{-1}\pi \left[ \hat{N}_j + \sum_{\alpha=1}^{n}(1-\hat{n}_{j,\alpha})\prod_{\substack{\rho=1 \\ \rho\neq\alpha}}^{n}\hat{n}_{j,\rho} \right]. \tag{62}$$

We show in appendix C that if $O_{F_j}^{(n)}$ is defined this way (*i.e.* symmetric for the exchange of any two fermionic operators and made out of number operators only), then it necessarily commutes with the operators $\bar{\sigma}_i$ and $\bar{\tau}_i$, both for $j \neq i$ (trivial) and for $j = i$. This implies that $O_F^{(n)}$ commutes with the Hamiltonian and so one can factorize the exponential in the partition function. In the end, for odd values of $n$ we find

$$\text{Tr}\, e^{-\beta(H_F^{(n)}+iO_F^{(n)})} = \text{Tr}_{\text{phys}}\, e^{-\beta(H_F^{(n)}+iO_F^{(n)})} = \text{Tr}_{\hat{N}_j=1}\, e^{-\beta H_F^{(n)}} \times (-1-1)^N = (-2)^N \,\text{Tr}_{\text{c.s.}}\, e^{-\beta H^{(n)}}. \tag{63}$$

Unfortunately this derivation cannot be extended to the case of even $n$. Indeed, if we consider $n = 2m$ and we impose the same constraints on $O_{F_j}$ as in the odd case, then Eq. (60) becomes

$$\sum_{\substack{k=0 \\ k\neq 1}}^{m-1}\binom{2m}{k}[e^{-i\eta_k} + e^{-i\eta_{2m-k}}] + \binom{2m}{m}e^{-i\eta_m} = 0. \tag{64}$$

In general, for arbitrary $n$ there is no set of $\eta_\ell$ satisfying this equation. Moreover, even for the cases where such a solution actually exists, it is cumbersome to derive it analytically. In that case, in order to find an operator $O_F^{(n)}$ for which Eq. (12) holds one would have to relinquish the symmetry constraint. This makes the search for the imaginary term as hard as for the first version of the mapping. Thus, the "symmetrized" version of the mapping with a doubled physical sector turns out to be advantageous with respect to the "asymmetric" one just when considering clock-spin models with odd order.

## 4.1 A special case: $n = 3$

As a concrete example, let us consider the case $n = 3$. The mapping is simply achieved by explicitly rewriting Eq. (55) and Eq. (56) for $n = 3$. Analogously to Subs. 3.1, we shift to the notation $a_j$, $b_j$, $c_j$ for the destruction fermionic operators at site $j$. Straightforward algebra yields:

$$\bar{\sigma}_j = a_j^\dagger a_j + \omega b_j^\dagger b_j + \omega^2 c_j^\dagger c_j - \omega a_j^\dagger a_j b_j^\dagger b_j + 2\omega a_j^\dagger a_j c_j^\dagger c_j - \omega b_j^\dagger b_j c_j^\dagger c_j, \tag{65}$$

$$\bar{\tau}_j = b_j^\dagger a_j + c_j^\dagger b_j + a_j^\dagger c_j, \tag{66}$$

with $\omega = e^{i\frac{2\pi}{3}}$. As already discussed in Sec. 2, these choices for $\bar{\sigma}_j$ and $\bar{\tau}_j$ ensure that they act on the physical states at site $j$, both with $\hat{N}_j = 1$ and with $\hat{N}_j = 2$, the same way as the original clock-spin operators $\sigma_j$ and $\tau_j$ act on the corresponding clock-spin states. Although

this is evident for the $\hat{N}_j = 1$ states, it may be worth working out the explicit calculation for the $\hat{N}_j = 2$ states

$$\bar{\sigma}_j |1,1,0\rangle_j = |1,1,0\rangle_j \,, \qquad\qquad \bar{\tau}_j |1,1,0\rangle_j = |1,0,1\rangle_j \,,$$
$$\bar{\sigma}_j |1,0,1\rangle_j = \omega |1,0,1\rangle_j \,, \qquad\qquad \bar{\tau}_j |1,0,1\rangle_j = |0,1,1\rangle_j \,,$$
$$\bar{\sigma}_j |0,1,1\rangle_j = \omega^2 |0,1,1\rangle_j \,, \qquad\qquad \bar{\tau}_j |0,1,1\rangle_j = |1,1,0\rangle_j \,.$$

At the same time we have that

$$\bar{\tau}_j |0,0,0\rangle_j = \bar{\tau}_j |1,1,1\rangle_j = 0 = \bar{\sigma}_j |0,0,0\rangle_j = \bar{\sigma}_j |1,1,1\rangle_j \,, \tag{67}$$

as requested for the unphysical states.

Finally, we need to determine the imaginary interaction term. According to Eq. (62), a possible choice for $O_{\mathrm{F}_j}^{(3)}$ is

$$O_{\mathrm{F}_j}^{(3)} = \frac{\pi}{\beta}[\hat{N}_j + a_j^\dagger a_j b_j^\dagger b_j + a_j^\dagger a_j c_j^\dagger c_j + b_j^\dagger b_j c_j^\dagger c_j - 3 a_j^\dagger a_j b_j^\dagger b_j c_j^\dagger c_j] \,. \tag{68}$$

Note that, however, just for the case $n = 3$ the desired result can also be achieved with a simpler imaginary operator. Following the derivation in [20] for the spin 1 case, using $O_{\mathrm{F}}^{(3)} = \frac{\pi \hat{N}}{3\beta}$, then we see that

$$O_{\mathrm{F}_j}^{(3)} |\hat{N}_j = 0\rangle = 0 \,,$$
$$O_{\mathrm{F}_j}^{(3)} |\hat{N}_j = 1\rangle = \frac{\pi}{3\beta} |\hat{N}_j = 1\rangle \,,$$
$$O_{\mathrm{F}_j}^{(3)} |\hat{N}_j = 2\rangle = \frac{2\pi}{3\beta} |\hat{N}_j = 2\rangle \,,$$
$$O_{\mathrm{F}_j}^{(3)} |\hat{N}_j = 3\rangle = \frac{\pi}{\beta} |\hat{N}_j = 3\rangle \,,$$

which satisfies Eq. (12)

$$\sum_{|\mathrm{unph}_j\rangle} \langle \mathrm{unph}_j | e^{-i\beta O_{\mathrm{F}_j}^{(3)}} |\mathrm{unph}_j\rangle = 1 + e^{-i\pi} = 0 \,. \tag{69}$$

Therefore, given that (up to a phase) the traces over the physical states with $\hat{N}_j = 1$ and $\hat{N}_j = 2$ lead to the very same result, we obtain

$$\mathrm{Tr}\, e^{-\beta(H_{\mathrm{F}}^{(3)} + i\pi \hat{N}/3\beta)} = \mathrm{Tr}_{\mathrm{phys}}\, e^{-\beta(H_{\mathrm{F}}^{(3)} + i\pi \hat{N}/3\beta)} = \mathrm{Tr}_{\hat{N}_j=1}\, e^{-\beta H_{\mathrm{F}}^{(3)}} \times [e^{-i\pi/3} + e^{-2\pi i/3}]^N \,. \tag{70}$$

In the end we have

$$\mathrm{Tr}\, e^{-\beta(H_{\mathrm{F}}^{(3)} + i\pi \hat{N}/3\beta)} = \left[-i\sqrt{3}\right]^N Z_{\mathrm{c.s.}} \,. \tag{71}$$

This result is especially relevant: given that the imaginary term is simply an addition to the chemical potential, it enters the fermionic Matsubara Green function [51] at the zeroth level

$$\mathcal{G}(\varepsilon, i\omega_{\mathrm{F}_\nu}) = \frac{1}{i\omega_{\mathrm{F}_\nu} - \varepsilon - i\pi/3\beta} \,, \tag{72}$$

with $\omega_{\mathrm{F}_\nu} = \frac{2\pi}{\beta}(\nu + 1/2)$ the fermionic Matsubara frequencies. As duly noted by Fedotov and Popov [20] this correction to the propagator corresponds to a redefinition of the Matsubara frequencies

$$\omega'_{\mathrm{F}_\nu} = \omega_{\mathrm{F}_\nu} - \frac{\pi}{3\beta} = \frac{2\pi}{\beta}(\nu + 1/2 - 1/6) = \frac{2\pi}{\beta}(\nu + 1/3) \,. \tag{73}$$

Thus, in this case, the introduction of the imaginary interaction term does not introduce new vertices in the diagrammatic perturbative expansion.

# 5  A mapping-inspired interesting fermionic model

The mapping presented in the previous sections was developed with the main goal of improving numerical calculations for clock-spin models, by making available the Wick theorem once the Hamiltonian is rewritten in fermionic language. However, as we anticipated in the introduction, it can also provide fruitful intuitions for conceiving fermionic models with exotic properties. To see why this may be the case, let us consider the zero-temperature limit. In this case the imaginary term, that is proportional to $\beta^{-1}$, disappears, so that the mapped fermionic Hamiltonian becomes Hermitian. This results in coinciding low energy spectra of both the clock-spin and the fermionic Hamiltonian and the fermionic model inherits all the ground-state physics of its corresponding clock-spin parent.

To provide an interesting explicit example, we consider the clock-spin model proposed in a recent paper by Hu and Watanabe [52], which comprises recently discovered unexpected even-odd effects [53–57]. The Hamiltonian of the model is

$$H = -\frac{1}{2} \sum_{j=1}^{N} \left[ (\sigma_j \sigma_{j+1} + \sigma_{j+1}^\dagger \sigma_j^\dagger) + g(\tau_j + \tau_j^\dagger) \right], \tag{74}$$

with a transverse field of strength $g > 0$ on a lattice with $N$ sites and periodic boundary conditions, so that $\sigma_{N+1} = \sigma_1$. We assume the clock-spin operators $\sigma$ and $\tau$ to be of order three, which implies $\omega = e^{i\frac{2\pi}{3}}$.

In Ref. [52], the authors show that for $g \ll 1$ this model gives rise to spontaneous symmetry breaking (SSB), though its features are drastically different for chains with an even or odd number of sites. In the even case, the model exhibits all the well known traits of SSB: the ground state is (three-fold) degenerate in the limit $g = 0$ and the energy splitting shrinks exponentially in the thermodynamic limit for $g \ll 1$. Moreover, upon the introduction of a symmetry breaking field $\varepsilon$ the thermodynamic limit ($N \to \infty$) and the vanishing field limit ($\varepsilon \to 0$) are found to be non commuting. On the other hand, the odd case is found to be quite peculiar: Although for odd $N$ the ground state is unique even in the ordered phase ($g \ll 1$), the two limits of the mean value of the order parameter are nonetheless non-commuting. The authors conclude that, since the thermodynamic properties should not depend on the system size or boundary conditions, the model with odd $N$ actually represents an example of SSB in the absence of exact symmetry or degeneracy.

Interestingly, our mapping allows to reproduce the physics of the model in a fermionic system (at zero temperature). To show this, we first have to pick one of the two mappings developed, and use it to map the Hamiltonian (74) into its fermionic counterpart. One could wonder about which version of the mapping is the most suitable for the present task. However, as long as we are only interested in the ground state properties of the model, we can greatly simplify the mapping and still land on a fermionic Hamiltonian with the same ground state as the ones obtained by an exact application of the mappings. Let us prove this point. The Hamiltonian obtained by applying the first and second version of the mapping would be respectively given by

$$\tilde{H}_{\mathrm{F}} = -\frac{1}{2} \sum_{j=1}^{N} \left[ (\tilde{\sigma}_j \tilde{\sigma}_{j+1} + \tilde{\sigma}_{j+1}^\dagger \tilde{\sigma}_j^\dagger) + g(\tilde{\tau}_j + \tilde{\tau}_j^\dagger) \right], \tag{75}$$

$$\bar{H}_{\mathrm{F}} = -\frac{1}{2} \sum_{j=1}^{N} \left[ (\bar{\sigma}_j \bar{\sigma}_{j+1} + \bar{\sigma}_{j+1}^\dagger \bar{\sigma}_j^\dagger) + g(\bar{\tau}_j + \bar{\tau}_j^\dagger) \right]. \tag{76}$$

Notice that although the two Hamiltonian are formally identical, the operators $\tilde{\sigma}_j$ and $\tilde{\tau}_j$ are given in Eq. (30) and Eq. (31) respectively, while $\bar{\sigma}_j$ and $\bar{\tau}_j$ are given in Eq. (65) and Eq. (66);

hence the two actually differ from each other. We will be interested in the ordered phase ($g \ll 1$), so we consider $g = 0$ for now. Moreover, we fix the total number of fermions in the chain to be equal to the number of sites $N$.

Given the Hamiltonian in Eq. (75) (Eq. (76)), for a pair of adjacent sites the energy is minimized when $\tilde{\sigma}_j \tilde{\sigma}_{j+1}$ ($\bar{\sigma}_j \bar{\sigma}_{j+1}$) assumes the value $+1$. Thus, if the state at either the site $j$ or $j + 1$ is one of the unphysical states, the action of $\tilde{\sigma}_j \tilde{\sigma}_{j+1}$ ($\bar{\sigma}_j \bar{\sigma}_{j+1}$) yields zero, and the energy is not minimized. The physical states for the first mapping are those with $\hat{N}_j = 1$ on each site; this is consistent with having fixed the total number of fermions to $N$. For the second mapping, as we discussed in full detail in Sec. 4, also the states with $\hat{N}_j = 2$ are labeled as physical. However, having the total number of fermions fixed to $N$ implies that for a physical state with $\hat{N}_j = 2$ at site $j$ there will necessarily be an unphysical state with $\hat{N}_{j'} = 0$ at some site $j'$: therefore such a configuration does not minimize the energy. In light of this, we can safely restrict to the subspace with $\hat{N}_j = 1$ at each site. Once this is done, both versions of the mapping reduce to the Abrikosov-like implementation, that we reported for the single-site in Eqs. (23) and (24)

$$\tilde{\sigma}_{A_j} = a_j^\dagger a_j + \omega b_j^\dagger b_j + \omega^2 c_j^\dagger c_j, \tag{77}$$

$$\tilde{\tau}_{A_j} = b_j^\dagger a_j + c_j^\dagger b_j + a_j^\dagger c_j. \tag{78}$$

The resulting fermionic Hamiltonian– that we denote as $H_F$ –is explicitly given by

$$
\begin{aligned}
H_F = -\frac{1}{2} \sum_{j=1}^{N} \Big[ &(2 a_j^\dagger a_j a_{j+1}^\dagger a_{j+1} + 2 b_j^\dagger b_j c_{j+1}^\dagger c_{j+1} + 2 c_j^\dagger c_j b_{j+1}^\dagger b_{j+1} \\
&- a_j^\dagger a_j b_{j+1}^\dagger b_{j+1} - a_j^\dagger a_j c_{j+1}^\dagger c_{j+1} - b_j^\dagger b_j a_{j+1}^\dagger a_{j+1} \\
&- c_j^\dagger c_j a_{j+1}^\dagger a_{j+1} - b_j^\dagger b_j b_{j+1}^\dagger b_{j+1} - c_j^\dagger c_j c_{j+1}^\dagger c_{j+1}) \\
&+ g(a_j^\dagger b_j + b_j^\dagger c_j + c_j^\dagger a_j + b_j^\dagger a_j + c_j^\dagger b_j + a_j^\dagger c_j) \Big].
\end{aligned}
\tag{79}
$$

Crucially, we observe that the Hamiltonian (79) is such that $[H_F, \hat{N}_j] = 0 \; \forall j$, implying that the time evolution preserves the occupation number at each site. This, together with the fact that the mapping in Eqs. (77) and (78) is exact and bijective upon restriction to the $\hat{N}_j = 1$ subspace, adds consistency to our simplification of the mapping.

It is easy to show that for $g = 0$, in the case of even $N$ there are three degenerate ground states, given by

$$|S_1\rangle = \prod_{j=1}^{N} a_j^\dagger |0\rangle, \qquad |S_2\rangle = \prod_{j=1}^{N/2} b_{2j-1}^\dagger c_{2j}^\dagger |0\rangle, \qquad |S_3\rangle = \prod_{j=1}^{N/2} c_{2j-1}^\dagger b_{2j}^\dagger |0\rangle, \tag{80}$$

while for odd $N$ there is a unique ground state, that is

$$|S_1\rangle' = \prod_{j=1}^{N} a_j^\dagger |0\rangle. \tag{81}$$

In both cases the excited states are gapped.

We now introduce a symmetry breaking field, following [52]

$$V(\varepsilon) = -\frac{1}{2} \varepsilon \left( \omega^{-1} \sum_{j=1}^{N} \tilde{\sigma}_{A_j}^{(-1)^{j-1}} + \text{h.c.} \right), \tag{82}$$

where the identification $\sigma_{A_j}^{-1} \equiv \sigma_{A_j}^\dagger$ – which is indeed valid upon restriction to the $\hat{N}_j = 1$ subspace –is assumed. In the even-$N$ case, the introduction of this term in the Hamiltonian (79)

clearly favours one of the three ground states ($|S_2\rangle$), so that the limits of the mean value of the order parameter on the ground state necessarily do not commute. Explicitly:

$$\lim_{N\to\infty}\lim_{\varepsilon\to 0^+}\langle\Psi_0(\varepsilon)|\frac{1}{N}\sum_{j=1}^{N}\tilde{\sigma}_{A_j}^{(-1)^{j-1}}|\Psi_0(\varepsilon)\rangle \neq \lim_{\varepsilon\to 0^+}\lim_{N\to\infty}\langle\Psi_0(\varepsilon)|\frac{1}{N}\sum_{j=1}^{N}\tilde{\sigma}_{A_j}^{(-1)^{j-1}}|\Psi_0(\varepsilon)\rangle\,, \quad (83)$$

where $|\Psi_0(\varepsilon)\rangle$ is the ground state of the Hamiltonian given by $H_F+V(\varepsilon)$. This can be easily seen in the $g\ll 1$ limit: in this special case, for $\varepsilon\to 0$ the ground state is a symmetric combination of $|S_1\rangle$, $|S_2\rangle$ and $|S_3\rangle$, so that the LHS of Eq. (83) goes to zero. The RHS on the other hand will give $\omega$ as a result, since for $g=0$ and $\varepsilon\neq 0$ the ground state is given by $|S_2\rangle$.

In the odd $N$ case, although the ground state is unique, still the two limits will differ. Indeed, if we assume that $g=0$ and we first take the $\varepsilon\to 0$ limit (LHS in Eq. (83)), we get that the mean value of the order parameter has to be computed over the $|S_1\rangle$ state, giving 1 as a result. Taking the $N\to\infty$ limit first (RHS of Eq. (83)), the state

$$|S_2\rangle' = \left(\prod_{j=1}^{(N-1)/2} b_{2j-1}^{\dagger}c_{2j}^{\dagger}\right)b_N^{\dagger}|0\rangle\,, \quad (84)$$

which for $\varepsilon=0$ is an excited state, becomes the lowest one in energy: the mean value of the order parameter over this state yields $\omega$ as a result. The explicit calculations are reported in appendix D. However, we remark that upon restriction to the subspace with $\hat{N}_j=1$– a choice which is consistent with the dynamical evolution of the system –our mapping becomes exact and bijective. Therefore all the discussions and results reported in [52] hold in our fermionic representation as well, and we refer the interested reader to the work of Hu and Watanabe for more details.

In the end, by virtue of our mapping, we were able to take off from a clock-spin model exhibiting SSB with non conventional features and land on a fermionic model manifesting similar behaviour. Interestingly, chains of interacting fermions like the one described by the Hamiltonian in Eq. (79) can be effectively simulated on quantum computers [58].

## 6 Discussion and outlook

In the present work, we have shown an exact mapping between clock-spin and fermionic partition functions. The mapping is based on an extension of the Fedotov-Popov theory to clock-spins. In particular, we have mapped a generic $n$-th order clock-spin model onto a fermionic counterpart in a local way: this is done by associating $n$ fermions to each lattice site and identifying a portion of the fermionic Hilbert-space, dubbed as physical, with the clock-spin Hilbert-space. The clock-spin operators are then mapped onto local combinations of the fermionic creation and destruction operators, in such a way that they act on the physical states just as the original clock-spin operators act on the corresponding clock-spin states, while yielding zero when acting on the other– unphysical –states. In order to avoid contributions from the unphysical states in the computation of the fermionic partition function, a suitable imaginary interaction term is added to the mapped fermionic Hamiltonian. Most notably, in particular for mimicking clock-spin models on quantum simulators, such imaginary interaction terms are not necessary if the simulator has a fixed number of fermions per site. We finally prove that the resulting fermionic non-Hermitian model has the same partition function as the original clock-spin one.

We have explicitly derived two distinct mappings, both sharing the general properties just recalled, but differing among each other for the choice of the physical subspace. We have

shown that the identification of the physical sector is a crucial step in the development of the mapping, with the two possibilities assessed leading to two mappings with quite different properties. In particular, for the version of the mapping with doubled physical subspace we have derived a general expression for the imaginary interaction term, valid for any odd clock-spin order. For both the versions we have investigated explicit examples, where, interestingly, in the case of clock-spins of order $n = 3$ with the second version of the mapping the imaginary interaction term can be taken simply proportional to the number operator.

These results are particularly relevant for numerical computation, since they allow the use of various numerical tools for computing fermionic correlation functions in the study of clock-spin models. Moreover, in one dimension, they allow to use the substantial literature about bosonization of fermionic systems to assess the low energy physics of clock-spin models. Finally, as we have shown in Sec. 5, the mapping can also be used to conceive new interesting fermionic models with peculiar features starting from their already known clock-spin counterparts, or *vice versa*.

As a perspective of our work, we point out that the mapping can be seen as a first step for a more ambitious result: it is well known that parafermionic models are related to clock-spin ones, into which they can be mapped via a non-local transformation akin to Jordan-Wigner. Up to date, a truly local mapping on the level of operators from parafermions to fermions has not been achieved yet, despite attempts in this direction [32]. The present work constitutes the first half of the bridge that needs to be built in order to locally connect parafermions to (non-Hermitian) fermions.

# Acknowledgments

**Funding information** N.T.Z. acknowledges the funding through the NextGenerationEu Curiosity Driven Project "Understanding even-odd criticality". C.F. acknowledges support from the European Research Council (ERC) under the European Union's Horizon 2020 research and innovation program (grant agreements No. 679722 and No. 101001902)

# A Computation of the trace

In this appendix we give a proof of Eq. (11). Let us first consider an operator $e^{(A+B)}$ with $A = \sum_{ij} A_1^i \otimes A_2^j$ and $B = \mathbb{I} \otimes B_2$, and a state $|s\rangle = |s_1\rangle \otimes |s_2\rangle$ such that $A_1^i |s_1\rangle = 0$ for every $i$. Then one has that

$$e^{(A+B)} |s\rangle = \sum_l \frac{(A+B)^l}{l!} |s\rangle = \sum_l \frac{B^l}{l!} |s\rangle = e^B |s\rangle \,. \tag{A.1}$$

Indeed, for any positive integer $m$,

$$AB^m |s\rangle = \sum_{ij} A_1^i \otimes A_2^j B_2^m |s_1\rangle \otimes |s_2\rangle = \sum_{ij} \underbrace{A_1^i |s_1\rangle}_{0} \otimes A_2^j B_2^m |s_2\rangle = 0 \,. \tag{A.2}$$

We now add a third operator, defined as $C = C_1 \otimes \mathbb{I}$, with the property $C_1 |s_1\rangle = \alpha_1 |s_1\rangle$. $C$ does not necessarily commute with $A$, but it commutes with $B$. Thus, any term of the Taylor expansion of $\exp(A+B+C)$ presenting the $B$ or $C$ operators at the right of an $A$ operator can be rewritten as some string of powers of $A$, $B$ and $C$, times a term $AB^m C^k$, for some integers $m$ and $k$. Moreover

$$AB^m C^k |s\rangle = \sum_{ij} A_1^i C_1^k |s_1\rangle \otimes A_2^j B_2^m |s_2\rangle = \alpha_1^k \sum_{ij} A_1^i |s_1\rangle \otimes A_2^j B_2^m |s_2\rangle = 0 \,. \tag{A.3}$$

In light of this, we have

$$e^{(A+B+C)}|s\rangle = \sum_l \frac{(A+B+C)^l}{l!}|s\rangle = \sum_l \frac{(B+C)^l}{l!}|s\rangle = e^{B+C}|s\rangle = e^B e^C|s\rangle\,, \tag{A.4}$$

where the last equivalence is due to the fact that $[B,C]=0$. As claimed in the main text, this proves Eq. (11). It is sufficient to make the following identifications in order to get the thesis: $|s\rangle = |\text{unph}_i\rangle \otimes_{j\neq i}|s_i\rangle$, $A = H_{F_i}$, $B = H'_{F_i} + O'_{F_i}$ and $C = O_{F_i}$.

## B  Derivation of the $O_F$ operator in the $n = 4$ case

We focus on deriving the form of the operator $O_F^{(4)}$ on a single site, being the extension to the whole lattice trivial. By doing so, we can omit the lattice site index. The most general operator that one can build satisfying the condition stated in the main text (which include symmetry under the exchange of fermionic operators) has the following form, at least up to a term proportional to the identity

$$\begin{aligned}
O_F^{(4)} = \frac{\pi}{\beta}\Big[ &\alpha(a^\dagger a + b^\dagger b + c^\dagger c + d^\dagger d) \\
&+ \eta(a^\dagger a b^\dagger b + a^\dagger a c^\dagger c + a^\dagger a d^\dagger d + b^\dagger b c^\dagger c + b^\dagger b d^\dagger d + c^\dagger c d^\dagger d) \\
&+ \gamma(a^\dagger a b^\dagger b c^\dagger c + a^\dagger a b^\dagger b d^\dagger d + a^\dagger a c^\dagger c d^\dagger d + b^\dagger b c^\dagger c d^\dagger d) \\
&+ \delta a^\dagger a b^\dagger b c^\dagger c d^\dagger d\Big]\,,
\end{aligned} \tag{B.1}$$

with $\alpha$, $\eta$, $\gamma$ and $\delta$ real coefficients. To determine the appropriate values for the parameters, it is useful to build a table that reports the eigenvalues of $O_F^{(4)}$ on each eigenstate of the number operator. Recall that, by construction, $O_F^{(4)}$ is degenerate over the subspaces with fixed $\hat{N}$.

From Tab. 1 it is straightforward to see that two of the (many) possible combinations of the parameters yielding the desired phases are

$$(\alpha,\eta,\gamma,\delta) = (1,1,0,0)\,, \tag{B.2}$$
$$(\alpha,\eta,\gamma,\delta) = (0,1,1,0)\,. \tag{B.3}$$

Table 1: Result of the action of the single-site $O_F^{(4)}$ operator on the unphysical states. The eigenvalue $\lambda_s$ is defined by $O_F|s\rangle = \lambda_s|s\rangle$, $|s\rangle$ being on each line the indicated unphysical state. On the last column are reported the desired phases, as discussed in the main text.

| | $|s\rangle$ | $\lambda_s$ | $\varphi$ |
|---|---|---|---|
| $\hat{N}=0$ | $|0,0,0,0\rangle$ | $0$ | $2k\pi$ |
| $\hat{N}=2$ | $|1,1,0,0\rangle,\ \ldots$ | $\frac{\pi}{\beta}[2\alpha+\eta]$ | $(2k+1)\pi$ |
| $\hat{N}=3$ | $|1,1,1,0\rangle,\ \ldots$ | $\frac{\pi}{\beta}[3\alpha+3\eta+\gamma]$ | $2k\pi$ |
| $\hat{N}=4$ | $|1,1,1,1\rangle$ | $\frac{\pi}{\beta}[4\alpha+6\eta+4\gamma+\delta]$ | $2k\pi$ |

## C  Proof of commutation

Let us consider the case in which $n$ is odd and we choose to pick as physical both the states with occupation number $\hat{N}_i = 1$ and $\hat{N}_i = n-1$. Once the imaginary potential is chosen in

such a way that the trace over the unphysical states does not give a contribution to the final result, one is left with having to compute

$$\mathrm{Tr}_{\mathrm{phys}}\, e^{-\beta\left(H_{\mathrm{F}}^{(n)}+iO_{\mathrm{F}}^{(n)}\right)},$$

where $|\mathrm{phys}\rangle = \otimes_{i=1}^{N} |\mathrm{phys}_i\rangle$. We now prove that if $O_{\mathrm{F}_i}^{(n)}$ is made of number operators at site $i$ only and, moreover, is symmetric under the exchange of any of them, then it commutes with the rest of the Hamiltonian. In particular, the operator in Eq. (62) satisfies these conditions.

The terms in $H_{\mathrm{F}}^{(n)}$ involving fermionic operators at site $i$ will be those coming from the fermionic counterparts of $\sigma_i$ and $\tau_i$. As can be seen from Eq. (55), $\bar{\sigma}_i$ only contains number operators at site $i$. Then, if $O_{\mathrm{F}_i}^{(n)}$ satisfies the above assumptions, it is trivial to conclude that

$$\left[\bar{\sigma}_i, O_{\mathrm{F}_i}^{(n)}\right] = 0. \tag{C.1}$$

On the other hand $\bar{\tau}_i$ (Eq. (56)) contains the terms $f_{i,\alpha+1}^{\dagger} f_{i,\alpha}$, that may not commute with $O_{\mathrm{F}_i}^{(n)}$. However this is not the case, thanks to the requirement that $O_{\mathrm{F}_i}^{(n)}$ be invariant under the exchange of any two fermionic operators. Indeed, given that

$$[f_{i,\alpha+1}^{\dagger} f_{i,\alpha}, f_{i,\alpha+1}^{\dagger} f_{i,\alpha+1}] = -f_{i,\alpha+1}^{\dagger} f_{i,\alpha}, \qquad [f_{i,\alpha+1}^{\dagger} f_{i,\alpha}, f_{i,\alpha}^{\dagger} f_{i,\alpha}] = +f_{i,\alpha+1}^{\dagger} f_{i,\alpha}, \tag{C.2}$$

if to any term in $O_{\mathrm{F}_i}^{(n)}$ containing the product of a certain combination of number operators at site $i$, among which there is $\hat{n}_{i,\alpha}$, corresponds a symmetric one where $\hat{n}_{i,\alpha}$ is replaced by $\hat{n}_{i,\alpha+1}$, then their contributions to the commutator with $f_{i,\alpha+1}^{\dagger} f_{i,\alpha}$ cancel out. If this happens for any $\alpha \in \{1, \ldots, n\}$, which is indeed the case if $O_{\mathrm{F}_i}^{(n)}$ is invariant under exchange of any two fermionic operators, then $O_{\mathrm{F}_i}^{(n)}$ actually commutes with $\bar{\tau}_i$ as defined in Eq. (54)

$$\left[\bar{\tau}_i, O_{\mathrm{F}_i}^{(n)}\right] = 0. \tag{C.3}$$

But then we have that

$$[H_{\mathrm{F}}, O_{\mathrm{F}}] = 0. \tag{C.4}$$

In light of this we can factorize $e^{-\beta(H_{\mathrm{F}}^{(n)}+iO_{\mathrm{F}}^{(n)})} = e^{-\beta H_{\mathrm{F}}^{(n)}} e^{-i\beta O_{\mathrm{F}}^{(n)}}$.

# D Explicit proof of SSB

We discuss in two separate subsections the even and odd $N$ cases.

## D.1 $N$ even

For $g = 0$, upon restriction to the $\hat{N}_j = 1$ subspace, the three degenerate ground states of the Hamiltonian in Eq. (79) are

$$|S_1\rangle = \prod_{j=1}^{N} a_j^{\dagger} |0\rangle, \qquad |S_2\rangle = \prod_{j=1}^{N/2} b_{2j-1}^{\dagger} c_{2j}^{\dagger} |0\rangle, \qquad |S_3\rangle = \prod_{j=1}^{N/2} c_{2j-1}^{\dagger} b_{2j}^{\dagger} |0\rangle. \tag{D.1}$$

One can readily check that the action of the Hamiltonian on these states gives $E_1 = E_2 = E_3 = -N$. Following [52], we introduce the order parameter

$$\hat{z} = \frac{1}{N} \sum_{j=1}^{N} \tilde{\sigma}_{\mathrm{A}_j}^{(-1)^{j-1}} = \frac{1}{N}(\tilde{\sigma}_{\mathrm{A}_1} + \tilde{\sigma}_{\mathrm{A}_2}^{\dagger} + \ldots \tilde{\sigma}_{\mathrm{A}_{N-1}} + \tilde{\sigma}_{\mathrm{A}_N}^{\dagger}), \tag{D.2}$$

and the symmetry breaking field

$$V(\varepsilon) = -\frac{1}{2}\varepsilon N(\omega^{-1}\hat{z} + \text{h.c.}),$$ (D.3)

which commutes with $H(g = 0)$. Then we have

$$\hat{z}\,|S_1\rangle = 1\,|S_1\rangle\,,$$ (D.4)

$$\hat{z}\,|S_2\rangle = \omega\,|S_2\rangle\,,$$ (D.5)

$$\hat{z}\,|S_3\rangle = \omega^2\,|S_3\rangle\,,$$ (D.6)

and, as a consequence

$$[H(g = 0) + V]\,|S_1\rangle = [-N - N\varepsilon\cos(\omega)]\,|S_1\rangle\,,$$ (D.7)

$$[H(g = 0) + V]\,|S_2\rangle = \left[-N - \frac{N\varepsilon}{2}(\omega^{-1}\omega + \text{h.c.})\right]|S_2\rangle = [-N - N\varepsilon]\,|S_2\rangle\,,$$ (D.8)

$$[H(g = 0) + V]\,|S_3\rangle = [-N - N\varepsilon\cos(\omega)]\,|S_3\rangle\,.$$ (D.9)

So the symmetry breaking (SB) field selects the ground state $|S_2\rangle$. Thus the large system size limit and vanishing field limit of the order parameter cannot commute. Explicitly,

$$\lim_{N\to\infty}\lim_{\varepsilon\to 0^+}\langle\Psi_0(\varepsilon)|\hat{z}|\Psi_0(\varepsilon)\rangle = 0\,,$$ (D.10)

$$\lim_{\varepsilon\to 0^+}\lim_{N\to\infty}\langle\Psi_0(\varepsilon)|\hat{z}|\Psi_0(\varepsilon)\rangle = \omega\,,$$ (D.11)

where $|\Psi_0(\varepsilon)\rangle$ is the ground state of the Hamiltonian given by $H_{\text{F}} + V(\varepsilon)$. The first limit is zero because for $\varepsilon = 0$ and $g \ll 1$ the ground state is a symmetric combination of the three above $(\Psi_0(0) \propto (|S_1\rangle + |S_2\rangle + |S_3\rangle))$, and $1 + \omega + \omega^2 = 0$ since $\omega = \exp(i2\pi/3)$. The second one is $\omega$ because for $\varepsilon \neq 0$ (and $g = 0$) the SB field selects the ground state $|S_2\rangle$.

## D.2  $N$ odd

For $g = 0$, upon restriction to the $\hat{N}_j = 1$ subspace the ground state is unique

$$|S_1\rangle' = \prod_{j=1}^{N} a_j^\dagger\,|0\rangle\,.$$ (D.12)

And its energy is $E_1' = -N$. The states corresponding to $|S_2\rangle$ and $|S_3\rangle$, that we denote here with a prime, are given by

$$|S_2\rangle' = \left(\prod_{j=1}^{(N-1)/2} b_{2j-1}^\dagger c_{2j}^\dagger\right)b_N^\dagger\,|0\rangle\,,\qquad |S_3\rangle' = \left(\prod_{j=1}^{(N-1)/2} c_{2j-1}^\dagger b_{2j}^\dagger\right)c_N^\dagger\,|0\rangle\,.$$ (D.13)

These are now excited states, with energy $E_2' = E_3' = -(N-1) + \frac{1}{2} = -N + \frac{3}{2}$. So the gap with respect to the ground state $|S_1\rangle'$ is $\Delta = \frac{3}{2}$. The order parameter is defined similarly to the even case

$$\hat{z} = \frac{1}{N}\sum_{j=1}^{N}\tilde{\sigma}_{A_j}^{(-1)^{j-1}} = \frac{1}{N}(\tilde{\sigma}_{A_1} + \tilde{\sigma}_{A_2}^\dagger + \ldots \tilde{\sigma}_{A_{N-2}} + \tilde{\sigma}_{A_{N-1}}^\dagger + \tilde{\sigma}_{A_N})\,,$$ (D.14)

so that again we get

$$\hat{z}\,|S_1\rangle' = 1\,|S_1\rangle'\,,$$ (D.15)

$$\hat{z}\,|S_2\rangle' = \omega\,|S_2\rangle'\,,$$ (D.16)

$$\hat{z}\,|S_3\rangle' = \omega^2\,|S_3\rangle'\,.$$ (D.17)

Then, introducing the same SB field as before

$$V(\varepsilon) = -\frac{1}{2}\varepsilon N(\omega^{-1}\hat{z} + \text{h.c.}),$$ (D.18)

one has

$$[H(g=0) + V]|S_1\rangle' = [-N - N\varepsilon\cos(\omega)]|S_1\rangle',$$ (D.19)

$$[H(g=0) + V]|S_2\rangle' = [-N + \frac{3}{2} - N\varepsilon]|S_2\rangle',$$ (D.20)

$$[H(g=0) + V]|S_3\rangle' = [-N + \frac{3}{2} - N\varepsilon\cos(\omega)]|S_3\rangle'.$$ (D.21)

Then $E_1'(\varepsilon) - E_2'(\varepsilon) = -\frac{3}{2} + N\varepsilon(1 - \cos\omega)$, which is positive for fixed $\varepsilon$ and $N \to \infty$. Then the SB field selects a different ground state in the odd case as well. Thus, the large system size limit and vanishing field limit of the order parameter cannot commute, despite the fact that the ground state is unique.

Explicitly,

$$\lim_{N\to\infty}\lim_{\varepsilon\to 0^+}\langle\Psi_0(\varepsilon)|\hat{z}|\Psi_0(\varepsilon)\rangle = 1,$$ (D.22)

$$\lim_{\varepsilon\to 0^+}\lim_{N\to\infty}\langle\Psi_0(\varepsilon)|\hat{z}|\Psi_0(\varepsilon)\rangle = \omega,$$ (D.23)

where $|\Psi_0(\varepsilon)\rangle$ is the ground state of the Hamiltonian given by $H_F + V(\varepsilon)$. The first limit is 1 because for $\varepsilon = 0$ (and $g = 0$) the ground state is $|S_1\rangle'$. The second one is $\omega$ because for $\varepsilon \neq 0$ (and $g = 0$) the SB field selects as ground state the state $|S_2\rangle'$.

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
