# Peer review of "An exact local mapping from clock-spins to fermions"

_SciPost Physics Core, doi:SciPost Phys. Core 6, 055 (2023)_

## Round 1 · Referee Report · Anonymous (Referee 1) · 2023-5-2

Strengths

  • It is an interesting mapping to a fermionic system of clock models, a thing that was unexpected.

Report

  • I do not see anything physically interesting. I mean, in the similar context of Z_2 spin chains, the Jordan-Wigner mapping was able to map an apparently unmanageable system (the quantum Ising chain in transverse field) to solvable quadratic fermions [P. Pfeuty, Annals of Physics 57, 79 (1970)]. In this representation the model showed Majorana fermions at the boundaries in the symmetry-broken phase [A. Y. Kitaev, Physics-Uspekhi 44, 131 (2001)], so that the magnetization transition in the spin representation was mapped to a topological transition in the fermionic one.

Nothing like that can I see here. The fermionic models that the authors obtain are not quadratic, so not solvable. Hence there is not the interesting point of making something complex simpler.

Is there at least some physical conclusion? For instance an interpretation of some Z_n symmetry breaking in terms of boundary topological modes in the fermionic representation (in analogy with the quantum Ising chain in transverse field)? What does their work add to the parafermions introduced in [1603.00095] in terms of physical interpretation and possibility to make analyses easier? The authors should answer to these questions before their work can be considered for publication in SciPost

Requested changes

In the present form the paper seems to me not suitable for publication in SciPost. The authors should consider some physical problem related to clock models (see ``Weak points'') and show that their method makes the analysis easier or gives new physical insight.

---

## Round 1 · Referee Report · Anonymous (Referee 2) · 2023-7-6

Strengths

1- The proposed mapping between general clock-spins and fermions is exact and local 2- The results of this work open to the possibility of employing existing techniques to tackle 1D interacting fermionic system to clock-spin models 3- Clock-spins enjoy unusual properties and are likely to rise to prominence as they can be realized in artificial systems or simulated at the quantum level

Weaknesses

1- The proposed mapping implies a substantial growth of the Hilbert space, since most of the fermionic states are unphysical. It is not clear from the paper's discussion how this affects the numerical implementation and efficiency of the mapping. 2- The author discuss a n=3 clock-spin model with unusual features and manage to reproduce those within the fermion language. It is a pity that this exercise is not used to draw some general conclusions or to present a deeper explanation of the peculiarities of the even/odd effects in this model.

Report

This manuscript details an exact, local mapping between clock-spins and fermions, developing a generalization of the Fedotov-Popov method to map spin-1/2 degrees of freedom into spinless fermions.
The paper is clearly written and motivated and its content seems correct to me. With the rise in interest toward synthetic materials, the understanding and exploitation of clock-spin degrees of freedom can become a growing field of activity for the community and thus I would favor publication of this work, once some minor revisions are implemented. However, at the moment I feel inclined in recommending its acceptance in a more specialized journal, such as SciPost Physics Core, since I am not sure that these results rise to the level of general interest aimed at by the flagship SciPost journal.

Requested changes

1- In the middle of page 2, not enough references are provided to vindicate the overwhelming amount of results for the quantum Ising chain. The authors should add additional citations, including some of the existing review articles. 2- Immediately after [10-12], the authors write that the non-locality of the Jordan-Wigner transformation has hindered its application to numerical approached in 1D. I feel that this sentence does not do justice to the great success of the JW when local interactions are considered. I think that the author should clarify that the issue arise when dealing with long-range interactions. 3- The authors sometime spell Fedetov and sometime Fedotov. Consistency should be restored through the manuscript. 4- After eq. (17) on page 6, it is written that a periodicity is implied in the number of species of fermions, so that f_{n+1} has to be identified with f_n. This is a crucial assumption for the whole construction which should be highlighted earlier in the construction of the mapping, preferably in Sec. 2. 5- I do not understand how the addition of that constant in eq. (43) does not affect the unphysical states as well, giving them a finite contribution to the partition function. The author should elaborate. 6- Before eq. (74) on page 16, the authors claim that they can drop the interaction piece of the Hamiltonian, since they are looking for the ground state only. Such statement is not correct in general and thus this sentence should be reformulated. 7- I find the solution and discussion in sec. 5 too rushed and I would like for the authors to add more details on the derivation and a discussion on what generates this weird even/odd effect in the fermionic language. It is clearly uncommon for the ground state degeneracy to depend on the parity of the chain length and even more for the order parameter to show such dependency...

---

## Round 2 · Author Response

Dear Editor,

We hereby resubmit our article "An exact local mapping from clock-spins to fermions" for resubmission to SciPost Physics Core.
We thank the two Reviewers for reading our manuscript and for the interesting points raised. Both of them do not have any concern about the validity of our results and see a point of novelty, and even of surprise (Referee 1 in "Strengths") in our results. Other than that, Referee 2 suggests publication on SciPost Physics Core after minor revision, while Referee 1 is more vague and suggests deeper changes in our manuscript in order to reconsider it for publication on Scipost. In both cases-if we understood correctly-the main criticism is that our mapping does not provide any explicit new insight in the physics of a specific system. We do agree on this point, but we think that even as it is our article contains interesting results. Indeed, in the original version of our manuscript, we identified a scheme to address clock-spin models by means of perturbative schemes enjoying Wick's theorem, and we developed a machinery to locally map clock-spin models to fermionic models with a corresponding translation of the physical properties. This second point-as also stated by Referee 2-might have an impact on synthetic material design. In the revised version, mainly motivated by Referee 1, we identify a third implication of our mapping: as long as one dimensional systems are considered, we developed a framework that opens the possibility of inspecting the low energy properties of clock spin models by using the well established tools of bosonization of interacting fermions. This point is relevant because it enlarges the consequences of our findings to the domain of analytical treatments of complex systems.

Based on these considerations, and on the point by point response we provide below, we hope that the Editor agrees with us that the article might be accepted to SciPost Physics Core in its present form, whose differences with respect to the previous version are reported in the attached list of changes.

On behalf of all the Authors,

Sincerely,

Simone Traverso
* * *
Point by point response
* * *
Referee 1:

"I do not see anything physically interesting. I mean, in the similar context of Z_2 spin chains, the Jordan-Wigner mapping was able to map an apparently unmanageable system (the quantum Ising chain in transverse field) to solvable quadratic fermions [P. Pfeuty, Annals of Physics 57, 79 (1970)]. "

We of course do agree that the Jordan-Wigner is more powerful than our transformation. Indeed, it represents a milestone of theoretical physics. However, we would like to point out that it is not possible to write a mapping from generic clock-spin models to free fermions since clock models are not necessarily integrable.

"In this representation the model showed Majorana fermions at the boundaries in the symmetry-broken phase [A. Y. Kitaev, Physics-Uspekhi 44, 131 (2001)], so that the magnetization transition in the spin representation was mapped to a topological transition in the fermionic one."

This clever identification by Kitaev is related to the non-local character of the Jordan-Wigner transformation. Indeed, going from local to topological order with a local transformation is not possible. Our mapping is in this sense more similar to the Holstein-Primakoff transformation.

"Nothing like that can I see here. The fermionic models that the authors obtain are not quadratic, so not solvable. Hence there is not the interesting point of making something complex simpler."

We thank the Referee for this observation, which forced us to think better about the implications of our mapping. Indeed, from the point of view of analytical treatments, our mapping allows for a rather striking simplification in the analysis of the low energy properties of clock-spin models: it makes the tools related to the bosonizaion of fermionic interacting one dimensional systems available.

"Is there at least some physical conclusion? For instance an interpretation of some Z_n symmetry breaking in terms of boundary topological modes in the fermionic representation (in analogy with the quantum Ising chain in transverse field)? What does their work add to the parafermions introduced in [1603.00095] in terms of physical interpretation and possibility to make analyses easier? The authors should answer to these questions before their work can be considered for publication in SciPost"

Aside from what we have stated before, a further physical conclusion is that we have proven that the physical properties of any clock model can be encoded into interacting fermionic systems, and moreover in a class of fermionic systems that are particularly suitable to be implemented on quantum computers.

Requested Changes:

"In the present form the paper seems to me not suitable for publication in SciPost. The authors should consider some physical problem related to clock models (see "Weak points") and show that their method makes the analysis easier or gives new physical insight."

While we followed the advice of the Referee about making a more clear statement on the significance of our results by adding a comment about the analytical tools it enables through bosonization, we decided to keep the structure of our article unchanged, without new references to specific models.
* * *
Referee 2:

"This manuscript details an exact, local mapping between clock-spins and fermions, developing a generalization of the Fedotov-Popov method to map spin-1/2 degrees of freedom into spinless fermions.
The paper is clearly written and motivated and its content seems correct to me. With the rise in interest toward synthetic materials, the understanding and exploitation of clock-spin degrees of freedom can become a growing field of activity for the community and thus I would favor publication of this work, once some minor revisions are implemented. "

We thank the Referee for the positive assessment of our work and for suggesting its publication after minor revision.

"However, at the moment I feel inclined in recommending its acceptance in a more specialized journal, such as SciPost Physics Core, since I am not sure that these results rise to the level of general interest aimed at by the flagship SciPost journal."

We agree with the Referee and hence we have resubmitted to SciPost Physics Core.

Requested changes:

"1- In the middle of page 2, not enough references are provided to vindicate the overwhelming amount of results for the quantum Ising chain. The authors should add additional citations, including some of the existing review articles. "

We have added the references.

"2- Immediately after [10-12], the authors write that the non-locality of the Jordan-Wigner transformation has hindered its application to numerical approached in 1D. I feel that this sentence does not do justice to the great success of the JW when local interactions are considered. I think that the author should clarify that the issue arise when dealing with long-range interactions."

We have clarified this point.

"3- The authors sometime spell Fedetov and sometime Fedotov. Consistency should be restored through the manuscript."

We have fulfilled the request.

"4- After eq. (17) on page 6, it is written that a periodicity is implied in the number of species of fermions, so that f_{n+1} has to be identified with f_n. This is a crucial assumption for the whole construction which should be highlighted earlier in the construction of the mapping, preferably in Sec. 2."

We have fulfilled the request.

"5- I do not understand how the addition of that constant in eq. (43) does not affect the unphysical states as well, giving them a finite contribution to the partition function. The author should elaborate."

We have added a sentence and an equation to clarify this point.

"6- Before eq. (74) on page 16, the authors claim that they can drop the interaction piece of the Hamiltonian, since they are looking for the ground state only. Such statement is not correct in general and thus this sentence should be reformulated."

We have completely rewritten the part of section 5 concerning the derivation of the fermionic Hamiltonian through our mapping.

"7- I find the solution and discussion in sec. 5 too rushed and I would like for the authors to add more details on the derivation and a discussion on what generates this weird even/odd effect in the fermionic language. It is clearly uncommon for the ground state degeneracy to depend on the parity of the chain length and even more for the order parameter to show such dependency…"

We have added an appendix with some explicit calculations concerning the results in section 5. We have also added a sentence to clarify that upon restriction to the $\hat{N}_j=1$ subspace our mapping becomes exact and bijective and therefore all the results and interpretations discussed in PRB 107, 195139 (2023) hold in our fermionic representation as well. For this reason, although we have added details to the derivation of the fermionic Hamiltonian as requested by the referee, we did not go into more details in discussing the results, preferring to refer the interested reader to the original work of Hu and Watanabe.

---

## Round 2 · List of Changes

1- In the abstract we have added a sentence about the fact that in 1D our mapping allows to use bosonization to access the low energy properties of clock-spin models.

2- At page 2 in the Introduction, just after we recall references [16-19], we have clarified the sentence about the complications that can occur in the application of the Jordan-Wigner transformation in presence of long range interactions.

3- At page 3, few lines before the description of the paper structure, we have added a sentence about the possibility of using bosonization in 1D. At page 3, in section 2, at the end of the paragraph below Eq. (3), we have added a sentence anticipating the formal periodicity in the flavor indexes, which are defined mod n (clock-spin order).

4- At page 9, in section 3.1, just after Eq. (43), we have added a sentence and an equation (Eq. (44)) clarifying the reason why the addition of a constant term to the operator O_F does not affect the trace over the unphysical states.

5- In section 5, from the fourth paragraph, starting with Interestingly, our mapping'', up to the one below Eq. (79), ending withsimplification of the mapping'', major changes have been made, with the aim of clarifying how the fermionic Hamiltonian is derived. A sentence has been added below Eq. (82) to clarify the form of the symmetry breaking field. Two sentences have been added right below Eq. (84), to clarify the relation of our mapped fermionic model with the original clock-spin model.

6- In the second-last paragraph of the conclusions we have added a sentence about the fact that in 1D our mapping allows to use bosonization to access the low energy properties of clock-spin models.

7- An entire appendix (D) has been added to explicit some of the calculations omitted for the sake of brevity in section 5.

8- We have updated the bibliography with several references, in particular about some major results concerning the Ising model and clock-spin models.

---

## Editorial Decision

published